# Defining and measuring bedtime routines in families with young children—A DELPHI process for reaching wider consensus

**George Kitsaras** [1]*, **Michaela Goodwin**[1], **Julia Allan**[2], **Iain A. Pretty**[1]

**1** Division of Dentistry, Dental Health Unit, University of Manchester, Manchester, United Kingdom,
**2** Institute of Health Sciences, University of Aberdeen, Aberdeen, United Kingdom

* Georgios.kitsaras@manchester.ac.uk

**Data Availability Statement:** All relevant data are within the manuscript and its Supporting information files.

## Abstract

### Introduction

Bedtime routines are one of the most common family activities. They affect children' wellbeing, development and health. Despite their importance, there is limited evidence and agreement on what constitutes an optimal bedtime routine. This study aims to reach expert consensus on a definition of optimal bedtime routines and to propose a measurement for bedtime routines.

### Method

Four-step DELPHI process completed between February and March 2020 with 59 experts from different scientific, health and social care backgrounds. The DELPHI process started with an expert discussion group and then continued with 3 formal DELPHI rounds during which different elements of the definition and measurement of bedtime routines were iteratively refined. The proposed measurement of bedtime routines was then validated against existing data following the end of the DELPHI process.

### Results

At the end of the four round DELPHI process and with a consistent 70% agreement level, a holistic definition of bedtime routines for families with young children between the ages of 2 and 8 years was achieved. Additionally, two approaches for measuring bedtime routines, one static (one-off) and one dynamic (over a 7-night period) are proposed following the end of the DELPHI process. A Bland-Altman difference plot was also calculated and visually examined showing agreement between the measurements that could allow them to be used interchangeably.

### Discussion

Both the definition and the proposed measurements of bedtime routines are an important, initial step towards capturing a behavioural determinant of important health and developmental outcomes in children.

**Funding:** This study forms part of a wider project funded by the Public Health Intervention Development Scheme of the Medical Research Council in the United Kingdom (ref.: MR/T002980/1).

**Competing interests:** The authors have declared that no competing interests exist.

## Introduction

Bedtime routines are amongst the most common family activities with virtually all families implementing a type of routine before children go to bed [1, 2]. Bedtime routines include a range of activities from brushing teeth to reading a book with the child [2–4]. Bedtime routines vary between families, but some core activities are consistently included. Bedtime routines can affect children's development, wellbeing and health as well as parental wellbeing and family functioning [2, 3, 5–7]. There is a growing recognition of the importance of bedtime routines yet there is little consensus on what constitutes an optimal bedtime routine. Different guidelines exist on different specific elements of a good bedtime routine such as oral hygiene practices before bed for young children [8] or recommended hours of sleep for children [9]. Recently, a systematic review [2] proposed different activities that should be considered during bedtime but fell short of providing a holistic definition for bedtime routines. Given the importance of bedtime routines, the lack of a clear, consensus-based definition of what constitutes an optimal bedtime routine limits health professionals' ability to communicate best practice effectively with families and prevents the scientific community from synthesising and further developing the empirical evidence base. Further work is therefore essential to clearly define this dynamic, repetitive family behaviour.

Apart from defining bedtime routines, another shortcoming can be found in measuring and quantifying them. The existing evidence on the importance of bedtime routines comes primarily from changes in specific targeted behaviours (e.g., toothbrushing) and subsequent improvements across specific metrics attached to those behaviours (e.g., dental health) rather than from the quality of the entire routine in a holistic fashion. Ideally, a robust method of measuring bedtime routines to quantify pre- and post-intervention changes is required to better understand the mechanisms involved in, how they affect children's development, wellbeing and health and identify opportunities to apply interventions to improve outcomes.

At present, one standardised measurement of bedtime routines, the Bedtime Routine Questionnaire (BRQ) [10] offers a validated approach to quantifying bedtime routines in families with young children. Despite its merits (provision of separate scores for weekdays and weekends and production of separate scales related to bedtime routines), the BRQ deploys a retrospective approach in assessing "typical" bedtime routines potentially limiting its utility. Bedtime routines are dynamic, are open to many environmental, social and personal influences every night and cover many different activities that ideally need to be consistently repeated each night to maximise their impact. As with observational data and diaries, biases including desirability bias, recall bias and rater fatigue can all affect the quality and quantity of data that can be obtained retrospectively [10, 11]. In order to effectively quantify full bedtime routines that accounts for differences between activities involved in such routines, differences between weekdays and weekends and to limit effect of biases, a new approach is necessary.

To both define and measure bedtime routines in families with young children a DELPHI process to reach consensus within a wide group of experts from different disciplines is proposed. The DELPHI method is a structured process where professionals with high levels of expertise in a given area communicate and share their professional opinions over multiple, iterative rounds until agreement is reached [12]. The DELPHI process is anonymous allowing for professionals to share their views without risk of social conformism [12]. At present, this process has not been used within bedtime routine research presenting a clear opportunity that needs to be explored.

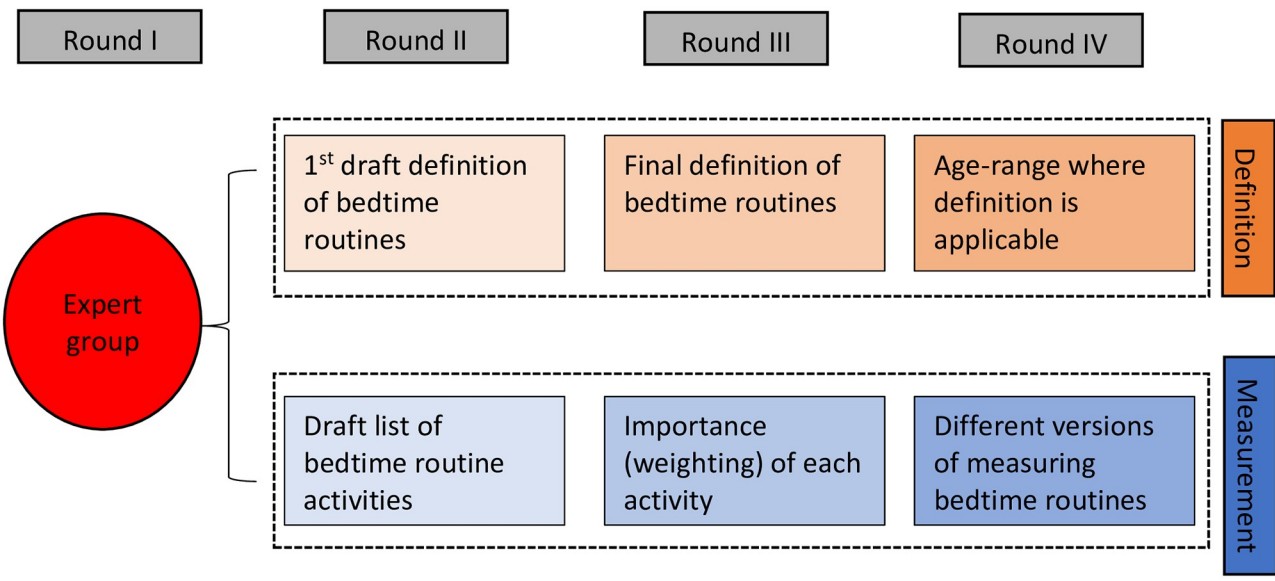

**Fig 1. DELPHI process.** Overview of DELPHI process rounds.

### Research aims

The two aims of the present study were to: (a) define optimal bedtime routines for families with young children and (b) propose an improved method of measuring bedtime routines in families with young children. Both aims were achieved with a holistic definition of bedtime routines for young children and a new measurement for bedtime routines provided.

### Methods

A four-round DELPHI process was initiated. Round I involved a full day expert group meeting where a preliminary definition of bedtime routines and the most important activities within bedtime were discussed. Rounds II, III and IV involved provision of anonymised feedback on a structured online questionnaire sent to experts who participated in the initial group as well as other professionals with relevant expertise in the area. Fig 1 illustrates the four-step DELPHI process. Experts provided consent in sharing their details as a requirement in participating in the expert group meeting.

### Inclusion criteria

For all steps of the DELPHI process inclusion criteria focused primarily on the academic and professional background of experts. Experts could come from different backgrounds including psychology, dentistry, medicine, public health, policy, education, nursing, midwifery, health visiting and sociology. For round I (expert group), invitations were targeted to experts to attend a full-day meeting in person. For rounds II, III and IV, no exclusion criteria were in place.

### Participants

In total, 59 experts participated across all four steps of the DELPHI process. Eleven experts took part in the expert group that started the DELPHI process followed by 25 experts in round

II, 20 experts in round III (80% retention) and finally, 13 experts in round IV (65% retention rate). (S1 Table) provides an overview of experts and (S2 Table) provides an overview of the questions asked to experts during the DELPHI rounds.

### Public and patient involvement

Due to the nature of this study, there was no active public and patient involvement. However, and based on previous work by the research team [3], parents of young children had expressed their views on the lack of a clear definition of what constitutes an optimal bedtime routine and the lack of clear guidance on this topic.

### Data analysis

Data analysis, from round II onwards, was performed using the count of endorsements (quantitative). For the definition of bedtime routines, experts could either agree or disagree with any element of the draft definition. If they disagreed, they were offered space for their recommendations and comments to be considered in the next step of the process. For measuring bedtime routines, during round II, items on the list of activities were endorsed as "important to achieve each night" and "less important". Items endorsed by more than 70% of participants were taken forward to the next round, as proposed by von der Gracht [12]. In round III and IV, participants provided a simple "agree" or "disagree" with proposed weights attached to each of the important bedtime routines activities and for the proposed, different approaches in measuring bedtime routines.

Following round IV, and once consensus was agreed on both aims, additional work was conducted around the proposed measurement of bedtime routines. Using an existing dataset from a previous study with parents and their children, the new measurements of bedtime routines were applied to examine the distribution and performance of the new measurements and scoring systems. This process involved a re-coding of participants' answers using the new scoring system. Comparisons between the proposed static and dynamic measurements of bedtime routines through the use of a Bland-Altman difference plot were calculated and examined to explore agreement [13].

## Results

### Definition of bedtime routines for young children

To define bedtime routines, firstly, the expert group discussed a working definition that encapsulated all important elements of an optimal bedtime routine considering best available evidence and advice. Following the expert group and during round II of the DELPHI process, experts were asked for their views on the proposed definition and to either agree or disagree and provide comments for improvements. Once changes were made to the definition, round III asked experts for their views again during which time consensus, over 70% agreement, was achieved. For the final, round IV, of the DELPHI process, experts were asked to provide their views on the age range of children where the definition of bedtime routines will be applicable. Due to a lack of 70% agreement in the first attempt, a subsequent, 2nd attempt was necessary to reach the required level of agreement as with all other rounds. Table 1 presents an overview of the process.

The final definition reached at the end of the DELPHI process is applicable for ages 2–8 and it includes best available advice for the bedtime activities related to children's health, wellbeing and development while considering parental wellbeing and the practicalities of implementing a bedtime routine in a busy, household setting.

**Table 1. Defining bedtime routines.**

| Round I Expert group | | |
|---|---|---|

Draft definition (drafted during the full day discussion meeting)
*"A routine should be formed around a calm environment and include some key behaviours including brushing teeth, having a bath/shower, reading book/sharing a book or storytelling, singing, praying, avoiding stimulating activities like television/tablets-mobiles/video-gaming before bed and avoid snacks/drinks before bed. These activities should be fairly consistent over the week including the weekend. Finally, children should go to bed at a reasonable time each night depending on their age."*

| Round II | | |
|---|---|---|
| Agreement with draft definition | Agreed | 10 |
| | Disagreed | 15 |

New definition
*"It is important to have a routine in place each night since a good bedtime routine can promote child health, development and wellbeing and allow parents some vital, free time each evening. A good bedtime routine should be formed around a calm environment and it should include different activities such as: (1) brushing teeth before bed, (2) avoiding snacks and drinks before bed except water and milk, (3) reading a book or sharing a book, telling a story, (4) avoiding stimulating activities such as television, mobile phones, tables and gaming consoles, (5) having a bath/shower but not necessarily every night and (6) interacting with the child in calm, relaxing activities such as playing together, cuddling, massaging and singing. All these activities should take place the hour before the child goes to bed and they should be fairly consistent across the week including the weekend. Inclusion of other calming, relaxing and interactive activities might be necessary based on family preferences. Finally, each night, children should go to bed early enough to allow them to sleep for the recommended age-appropriate time before they have to get up in the morning and for a minimum of 8 hours each night."*

| Round III | | |
|---|---|---|
| Agreement with draft definition | Agreed | 18 |
| | Disagreed | 2 |

Final definition
*"It is important to have a routine in place each night. A good bedtime routine can promote child health, development and wellbeing. Bedtime routines should be formed around a calm environment and include different activities such as: (1) brushing teeth right before going to bed (for children under 7, parents should actively brush children's teeth), (2) avoiding snacks and drinks after brushing teeth & limiting snacks and drinks the hour before bed, (3) reading or sharing a book or telling a story before bed, (4) avoiding stimulating activities such as television, mobile phones, tables and gaming consoles, and (5) interacting with the child in calm, relaxing activities such as playing together, cuddling, singing and/or having a bath/shower but not necessarily every night. All these activities should take place during the hour before the child goes to bed and they should be fairly consistent across the week including the weekend. Finally, each night, children should go to bed early enough to allow them to sleep for the recommended, age-appropriate time before they have to get up in the morning and for a minimum of 8 hours each night."*

| Round IV | | |
|---|---|---|
| | 1st Attempt | Expert preference |
| Proposed age range for definition | 1–7 years | 1 |
| | 1–8 years | 1 |
| | 2–7 years | 3 |
| | 2–8 years | 8 |
| | 2nd attempt | Expert preference |
| | 1–7 years | 0 |
| | 1–8 years | 0 |
| | 2–7 years | 3 |
| | 2–8 years | 10 |

## Definition of an optimal bedtime routine for children age 2–8

*"It is important to have a routine in place every night. A good bedtime routine can promote child health, development and wellbeing. Bedtime routines should be formed around a calm environment and include different activities such as: (1) brushing teeth before going to bed for 2 minutes using a fluoridated toothpaste(for children under 7, parents should actively brush children's*

*teeth), (2) avoiding snacks and drinks after brushing teeth and generally limiting snacks and drinks the hour before bed (water and unflavoured milk aside), (3) reading or sharing a book with children or simply telling a story before bed, (4) avoiding stimulating activities and electronic devices such as television, mobile phones, tables and gaming consoles, and (5) interacting with the child in calm, relaxing activities such as playing together, cuddling, singing and/or having a bath/shower but not necessarily every night. All these activities should take place the hour before the child goes to bed and they should be fairly consistent across the week and the weekend. Finally, each night, children should go to bed early enough to allow them to sleep for the recommended, age-appropriate time before they have to get up in the morning."*

## Measurement of bedtime routines: Two approaches (static/one-off and dynamic/repeated over a week) with weighting of options

For the measurement of bedtime routines, the DELPHI process started by compiling a list of relevant bedtime routine activities during the expert group (round I) some initial screening of the list was undertaken during the face-to-face meeting to condense the options before proceeding to the next DELPHI round. That list was shared with experts in round II to prioritise which activities were more or less important to include as part of an optimal routine. In round III experts were asked to assign weights on each activity for the purpose of producing a measure. Each expert was allocated 100 points to assign to the list of activities to indicate relative importance. Each expert could allocate all scores in one activity or spread them according to which activities were more/less important. Additionally, for that round, experts needed to consider need for consistency: how important each activity is to achieve every night and which activities are less important on a nightly basis. Finally, in round IV, experts were asked for their views on a one-off (static) and a 7-night (dynamic) measurement for bedtime routines. For that final round, experts needed to state their preference for the two different measurements. Table 2 presents the results of this process.

## A new approach in measuring bedtime routines: The BTR-Index

Following the end of the DELPHI process, a new approach in measuring bedtime routines, the BTR-Index is proposed. The proposed measurement index of bedtime routines for families with young children includes two versions; a one-off, static measurement (BTR-Index (S)) where parents receive a score out of 100 (0% no routine in place, under 50% sub-optimal bedtime routine, 100% excellent bedtime routine) based on a list of 6 activities weighted for their importance and a 7-night, dynamic measurement (BTR-Index (D)) where parents receive a score out of 100 based on which activities they complete over a week.

The 6 core activities and their respective scores based on their importance are: (a) brushing teeth before bed– 35 points, (b) time consistency for going to bed– 20 points, (c) book reading before bed– 15 points, (d) avoiding food/drinks before bed– 10 points, (e) avoiding use of electronic devices before bed– 10 points and (f) calming activities with child before bed including bath/shower, signing, talking etc.– 10 points. If a parent achieves all 6 areas as part of his/her bedtime routine, then they will receive a score of 100%, if they omit one or more of the activities, they will lose those points resulting in a lower overall score (for example, if they omit book reading/sharing a story before bed (-15 points) + if they allow use of electronic devices (-10 points) then the overall score will be 75%). The same scoring system is used for the dynamic measurement where depending how many nights a week parents achieve these activities they receive different, weighted scores multiplied by 1.0 if they achieve the activity at least 6 (6–7) nights a week, 0.7 if they achieve the activity at least 4 (4–5) nights a week, 0.5 if they

**Table 2. Measuring bedtime routines.**

| Round II | | | |
|---|---|---|---|
| **List of activities** | **Important to achieve every night (N = expert opinion)** | **Less important to achieve every night (N = expert opinion)** | **% Agreement that activity is important** |
| **Brushing teeth before bed** | 25 | 0 | 100% |
| **Food/drinks before bed** | 18 | 7 | 72% |
| **Avoiding use of electronic devices before bed** | 18 | 7 | 72% |
| **Reading/sharing a book/ story before bed** | 20 | 5 | 80% |
| **Consistency for time going to bed** | 21 | 4 | 84% |
| **Bath/shower before bed** | 3 | 22 | 12% |
| **Interactive activities with child before bed** | 15 | 10 | 60% |
| **Round III** | | | |
| **Weighting activities** | **Activity** | **M (SD) out of 100** | |
| | Brushing teeth | M = 35 (SD = 9.5) | |
| | Food/drinks before bed | M = 10 (SD = 5) | |
| | Avoiding use of electronic devices before bed | M = 10 (SD = 2.36) | |
| | Reading/sharing a book/story before bed | M = 15 (SD = 3.5) | |
| | Consistency for time going to bed | M = 20 (SD = 3.00) | |
| | Calming activities with child prior to bed | M = 10 (SD = 2.91) | |
| **Weighting consistency** | **Options** | **Experts' preference (N = 20)** | |
| | (A) Multiple scores by 1.0 if achieved 6–7 nights, 0.7 if achieved 4–5 nights, 0.5 if achieved 2–3 nights, 0.3 if achieved 1–2 nights and 0.1 if not achieved | 15 | |
| | (B) Multiple scores by 1.0 if achieved every night, 0.9 if achieved 6 nights, 0.7 if achieved 5 nights, 0.5 if achieved 4 nights, 0.3 if achieved 3 nights, 0.1 if achieved 1–2 nights and 0.0 if not achieved | 5 | |
| | (C) Add each night's scores and simply divide by 7 to achieve average score | 0 | |
| **Round IV** | | | |
| Preference for static vs. dynamic measurement or both | Static (one off) | 1 | |
| | Dynamic (multi-night) | 2 | |
| | Both | 10 | |
| | Neither | 0 | |

achieve the activity at least 2 (2–3) nights a week, 0.3 if they achieve the activity at least 1 night a week (1–2) nights and 0.1 if they don't achieve the activity at all during the week.

This proposed measurement can be used in different iterations from traditional paper-based to fully digital and electronic data collection tools. It is proposed that researchers adapt the method of data collection to better suit their research needs and the ever-changing research and societal landscape. (S3 Table) provides a summary of the 6 core activities that need to be covered as part of this proposed new measurement of bedtime routines.

## Validation of bedtime routines measurement

Using an existing dataset from a previous study [3] on bedtime routines for families with young children (n = 27), the new scoring systems were applied to the data to ensure that they

produced differences in bedtime routine scores across participants with optimal and sub-optimal routines. Within this existing dataset, parents completed a series of real time assessments of their bedtime routines over a 7-night period as part of their participation in a study conducted by the same research team. Both the one-off, static measurement of bedtime routines and the dynamic, 7-night measurement of bedtime routines were tested against the data already provided by these participants. For the static measurement, participants received up to 100 points based on the scoring system described for each bedtime routine activity. For the dynamic measurement, participants received scores based on how frequently they achieved each activity over the 7-night period again based on the proposed scoring and weighting system.

For example, in the static measurement, if a parent provided the following data:

$$Brushing + Book\ reading + Avoiding\ electronic\ devices = 35 + 15 + 10 = 60/100$$

For the dynamic measurement, if a parent provided the following data over a 7-night period, then:

$$\left( Brushing\ \frac{6}{7}\,nights \right) + \left( Book\ reading\ \frac{3}{7}\,nights \right) + \left( Avoiding\ electronic\ devices\frac{4}{7}\,nights \right)$$

$$+ \left( Food\ \&\ drinks\ before\ bed\ \frac{7}{7}\,nights \right) + \left( Time\ off\ to\ bed\ \frac{6}{7}\,nights \right)$$

$$+ \left( calming\ activities\frac{5}{7}\,nights \right)$$

$$= (35x1) + (15x0.5) + (10x0.7) + (10x1) + (20x1) + (10x0.7)$$

$$= 35 + 7.5 + 7 + 10 + 20 + 7 = 86.5/100$$

Scatter plots on the scores calculated for both the static and the dynamic assessment can be seen in Fig 2 below. A Bland-Altman difference plot was also calculated and visually examined for agreement between the measurements that could allow them to be used interchangeably. For the Bland-Altman plot, bias differences and mean values per individual scores between the static and dynamic measurements were calculated. Bias, standard deviation for differences in scores, lower level of agreement (mean difference -1.96 SD of differences) (LOA) and upper level of agreement mean difference +1.96 SD of differences) were also calculated in order to produce the plot. Fig 2 presents the result of the calculations. Interpretation considers the 95% confidence interval of the LoA, if these limits do not exceed the maximum allowed difference between methods, the two methods are considered to be in agreement and may be used interchangeably.

Both measures resulted in a wide range of scores that related meaningfully to identified differences in the quality of routines. Parents who scored highly on both measurements showed consistently optimal bedtime routines for example participant 26 scoring 100 in the static and 90 in the dynamic measurement and participant scoring 100 in the dynamic and 86.5 in the static measurement. On the contrary, participants with low bedtime routine scores in the static measurement showed low bedtime routine scores in the dynamic measurement for example participant 2 scoring 41 in the static and 35 in the dynamic measurement. Bias was calculated at 3.09, lower LOA at -18.01 and upper LOA at 32.20. Based on inspection, both methods did not exceed the allowed difference between methods and could therefore be used interchangeable for assessing bedtime routines in families with young children.

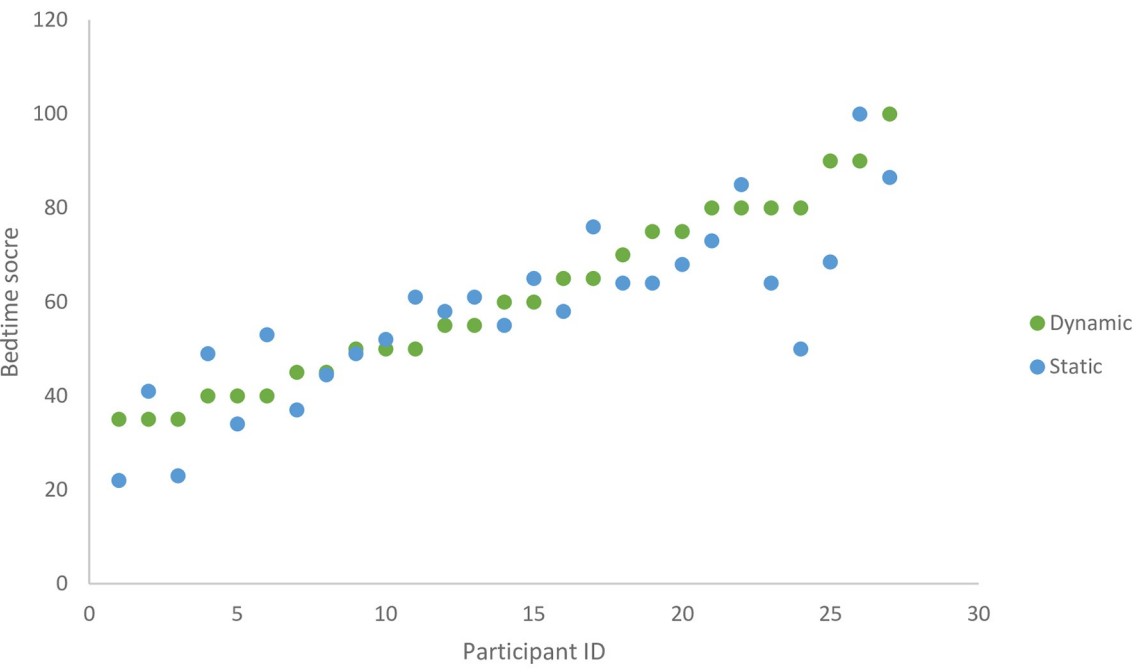

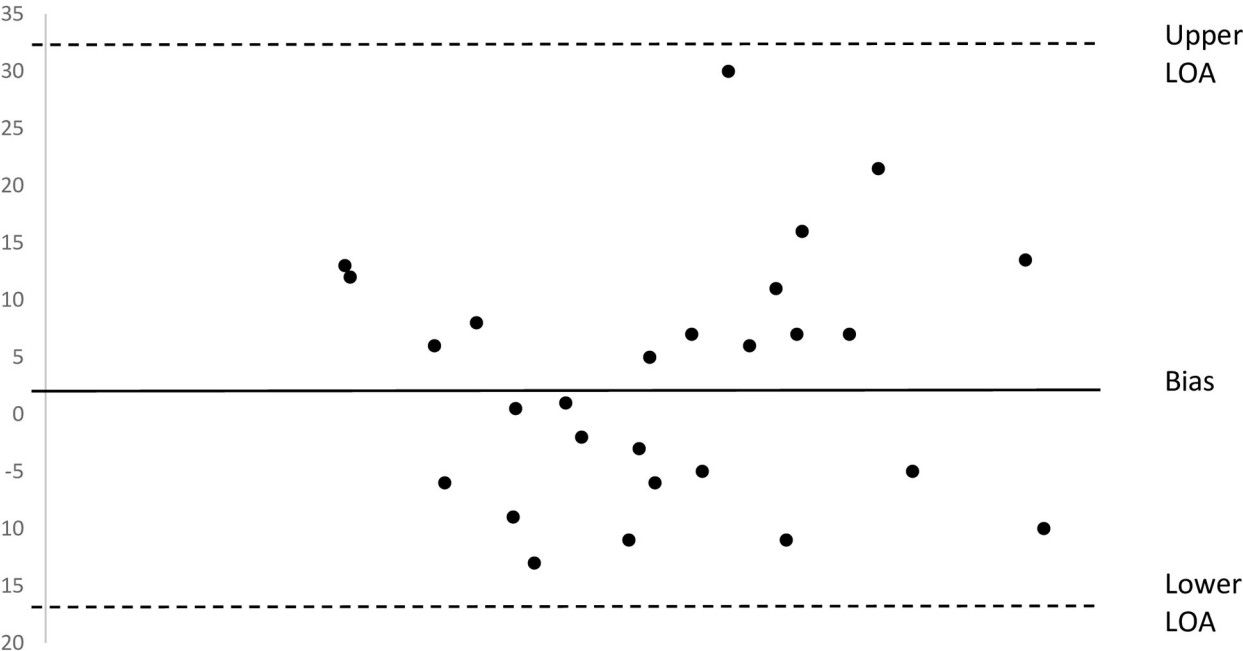

**Fig 2. Static vs. dynamic assessment of bedtime routines; differences in individual scores per type of measurement & Bland-Altman difference plot.**

## Discussion

This is the first study to generate a formal consensus from a group of experts with respect to the definition and measurement of optimal BTRs. Given the lack of a consistent approach to defining and measuring bedtime routines to date, consensus is vital to provide the foundation upon which effectiveness research can be built. Through this DELPHI process, both initial research aims were achieved: a holistic definition for bedtime routines for families with young children was proposed and agreed, and a method of measuring bedtime routines was developed and validated against existing data; the BTR Index.

The definition considers the parental stresses and difficulties that might arise at bedtime while incorporating best practice and available scientific advice about the content of an optimal bedtime routine. The language of the definition has, intentionally, been kept accessible to lay readers to ensure that advice can be easily absorbed by those who implement bedtime routines on a daily basis. Effective scientific communication is vital for all disciplines, as only through effective communication can the wider public make sense of important and, at times conflicting, messages [14]. With multiple sources of information available to parents from peer support groups to books and grey literature, there are a multitude of resources at hand when people seek ways of establishing and managing good bedtime routines. What parents currently lack is a robust yet comprehensible definition of what an optimal routine is to untie the complex signals and messages they receive. The definition of bedtime routines created in the current study goes a long way towards addressing this problem, effectively communicating what an optimal routine looks like.

In addition, a measurement for bedtime routines that utilises a dual, static and dynamic approach, has been developed to reflect the research need to be able to accurately capture the dynamic and fluid nature of bedtime routines. With both measurements producing similar results, there is added flexibility for researchers moving forward who might wish to opt for the faster, one-off rather than the more time-consuming dynamic measurement depending on the scope of their research project. Also, the static measure can be used as a checklist when assessing potential participants in studies around bedtime routines. The 7-night span of the dynamic measurement allows for observations regarding weekend and weekday effects on bedtime routines to be observed, something that could be missed with the static measurement. Also, the dynamic nature of the assessment could produce a more detailed picture of bedtime routines in families when compared to retrospective assessments.

### Limitations

This process and subsequent results have some limitations. The definition remains deliberately broad, and as a consequence does not consider the specific bedtime routine requirements of children with learning disabilities, health conditions and/or children in care. As for the proposed measurement of bedtime routines, one limitation to be highlighted is the lack of more robust validation work with data collection from a new sample, specific to testing and validating these proposed measures and/or formal comparison with existing measures. Additional work will be required to examine the structural validity and sensitivity of these measures similar to the robust work undertaken during the development of the BRQ [10]. Also, there was an expert retention loss during the four round DELPHI process that could have led to some alternative voices and opinions missed from the final definition and proposed measurement. To counter lost retention, we provided experts with sufficient time to comment and provide feedback however, for a limited number of experts, that was not sufficient resulting in lost retention. For the proposed index, further explorations will need to be in place moving forward to fully examine different cut-off points and determine where a routine seizes being beneficial

and optimal and slips into a suboptimal, poor routine. Finally, the current use of activity and consistency weightings will need further exploration to determine whether it is the best method of capturing these elements of bedtime routines.

## Conclusion

This DELPHI study and its outputs are an initial, yet important step in defining and quantifying bedtime routines. Both the proposed definition and measurement are preliminary and will need further validation work before they can be widely adopted. Nevertheless, this work through the engagement of a wide pool of experts can act as an important trigger for further scientific enquiries into a crucial set of behaviours that affect children's wellbeing and development.

## Supporting information

**S1 Table. Characteristics of experts participating in DELPHI process.**
(DOCX)

**S2 Table. Questions asked to experts as part of the DEPHI process.**
(DOCX)

**S3 Table. BTR index.**
(DOCX)

## Author Contributions

**Conceptualization:** George Kitsaras, Michaela Goodwin, Julia Allan, Iain A. Pretty.

**Formal analysis:** George Kitsaras, Michaela Goodwin.

**Funding acquisition:** Julia Allan, Iain A. Pretty.

**Investigation:** George Kitsaras.

**Methodology:** George Kitsaras, Julia Allan, Iain A. Pretty.

**Project administration:** George Kitsaras.

**Resources:** George Kitsaras.

**Supervision:** Julia Allan, Iain A. Pretty.

**Validation:** Iain A. Pretty.

**Writing – original draft:** George Kitsaras.

**Writing – review & editing:** George Kitsaras, Michaela Goodwin, Julia Allan, Iain A. Pretty.

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
