## [Decision Letter · Decision Letter 0]

18 Dec 2020

PONE-D-20-26667

Defining and measuring bedtime routines in families with young children; a DELPHI process for reaching wider consensus

PLOS ONE

Dear Dr. Kitsaras,

Thank you for submitting your manuscript to PLOS ONE. After careful consideration, we feel that it has merit but does not fully meet PLOS ONE’s publication criteria as it currently stands. Therefore, we invite you to submit a revised version of the manuscript that addresses the points raised during the review process.

 Considering my own reading of the paper and reviewer suggestions, I am recommending a major revision. If your willing to address the reviewer suggestion, we would like to consider the revised version of this paper. 

We look forward to receiving your revised manuscript.

Kind regards,

Srinivas Goli, Ph.D.

Academic Editor

PLOS ONE

Journal Requirements:

2. In your Methods section, please state whether you obtained consent from experts to publish their names prior to their engagement in this research.

5. Please upload a copy of Figure 3, to which you refer in your text on page 14. If the figure is no longer to be included as part of the submission please remove all reference to it within the text.

Additional Editor Comments (if provided):

Considering my own reading of the paper and reviewer suggestions, I am recommending a major revision. If your willing to address the reviewer suggestion, we would like to consider the revised version of this paper.

Reviewers' comments:

Reviewer's Responses to Questions

**Comments to the Author**

1. Is the manuscript technically sound, and do the data support the conclusions?

Reviewer #1: Yes

Reviewer #2: Yes

Reviewer #3: Partly

Reviewer #4: Yes

2. Has the statistical analysis been performed appropriately and rigorously? 

Reviewer #1: Yes

Reviewer #2: Yes

Reviewer #3: No

Reviewer #4: Yes

3. Have the authors made all data underlying the findings in their manuscript fully available?

Reviewer #1: Yes

Reviewer #2: Yes

Reviewer #3: No

Reviewer #4: Yes

4. Is the manuscript presented in an intelligible fashion and written in standard English?

Reviewer #1: Yes

Reviewer #2: Yes

Reviewer #3: Yes

Reviewer #4: Yes

5. Review Comments to the Author

Reviewer #1: Dear authors,

Thank you for your submission. This manuscript was very well written. I have, however, a couple questions/comments.

- Did you carry out a structured review to present existing evidence to the expert panel in Round 1? If not, please address this limitation of not having done so and if you did, please include this detail in your manuscript.

- It seems that you lost expert retention for subsequent rounds. Please address this address this limitation and what it means for your results.

- I am assuming you used 70% consensus agreement for all rounds? Is this correct? If not, please state what level you used for each round. And if you selected varying consensus levels for each round, please provide your reasoning as it is quite unconventional. If you did use 70% consensus, in Round 4, you accepted the proposed age range of 2-8 years, but you only have 8/13 = 62%.

- You propose a new index for measuring BTR from 0-100. Please clarify what your index scores indicates? (e.g., 0 = Poor; 100 = Excellent). Do you have a cutoff for "acceptable?" (e.g. 70)

- Can you clarify, did all 27 families completed both a static and dynamic questionnaire? I am thinking so, but I just wanted to make sure.

Reviewer #2: This article is a new initiation for reaching a consensus regarding bed time routines including oral hygiene and sleep in children and also, this article may serve as a basis for other similar studies in future.

There are some issues which need to be addressed:

• The children’s age range from 2 to 8 years is a wide variation considering the developmental milestones, for instance, preschool and school going children, sleep during daytime in preschool children hampering the sleep at night.

• “A previous study” mentioned in the validation of bedtime routine measurement section needs to be clarified.

• Minor grammatical errors such as incomplete first two sentences of methods in abstract section and so on.

• Uniformity in citing the references as per the journal requirement and complete citation of reference number 8.

Reviewer #3: 1) Abstract does not reflect the results of the manuscript.

2) What does the 1 1 3 and 8 mean in "Table 1 of the results section (Round IV)"

3) It will be good to mention what is meant by static and dynamic measurement in the section of "measurement of bedtime routines: two approaches with weighting of options"

4) Weighting consistency requires a bit more of explanation of how the scores are assigned

5) How are the points assigned to the 6 activities in the new approach. Is it based on previous studies or is it the authors description?

6) No information about the existing dataset used in the study.

7) How can we say that the validation is robust. To whom are the static and dynamic scores, as obtained from the new method, compared to?

8) Which additional variables have been used to establish this new BTR index

Reviewer #4: The research has been done on an unexplored area, which is a great work.

6. PLOS authors have the option to publish the peer review history of their article (what does this mean?). If published, this will include your full peer review and any attached files.

Reviewer #1: **Yes: **Dr. Breda Eubank

Reviewer #2: No

Reviewer #3: No

Reviewer #4: No

---

## [Author Response · Author response to Decision Letter 0]

13 Jan 2021

We have provided a detailed response to each reviewers' comment as an additional file with our submission. We hope our responses and subsequent actions will be sufficient.

---

## [Decision Letter · Decision Letter 1]

9 Feb 2021

Defining and measuring bedtime routines in families with young children; a DELPHI process for reaching wider consensus

PONE-D-20-26667R1

Dear Dr. Kitsaras,

We’re pleased to inform you that your manuscript has been judged scientifically suitable for publication and will be formally accepted for publication once it meets all outstanding technical requirements.

Kind regards,

Srinivas Goli, Ph.D.

Academic Editor

PLOS ONE

Additional Editor Comments (optional):

Satisfied with the revisions and recommending the paper for publication.

Reviewers' comments:

Reviewer's Responses to Questions

**Comments to the Author**

1. If the authors have adequately addressed your comments raised in a previous round of review and you feel that this manuscript is now acceptable for publication, you may indicate that here to bypass the “Comments to the Author” section, enter your conflict of interest statement in the “Confidential to Editor” section, and submit your "Accept" recommendation.

Reviewer #1: All comments have been addressed

Reviewer #2: All comments have been addressed

2. Is the manuscript technically sound, and do the data support the conclusions?

Reviewer #1: Yes

Reviewer #2: Yes

3. Has the statistical analysis been performed appropriately and rigorously? 

Reviewer #1: Yes

Reviewer #2: Yes

4. Have the authors made all data underlying the findings in their manuscript fully available?

Reviewer #1: Yes

Reviewer #2: Yes

5. Is the manuscript presented in an intelligible fashion and written in standard English?

Reviewer #1: Yes

Reviewer #2: Yes

6. Review Comments to the Author

Reviewer #1: Thank you for addressing all reviewers comments. I think your paper reads very well. Even though the results of your scoping review have not been published, I still think your paper would benefit by adding a statement that a scoping and systematic review were completed to establish evidence for your Delphi process.

Reviewer #2: The issues that had been previously raised are addressed in the revised version of the manuscript and this article will be of importance for similar studies in future.

7. PLOS authors have the option to publish the peer review history of their article (what does this mean?). If published, this will include your full peer review and any attached files.

Reviewer #1: **Yes: **Dr. Breda H.F. Eubank

Reviewer #2: No

---

## [Editor Report · Acceptance letter]

12 Feb 2021

PONE-D-20-26667R1 

Defining and measuring bedtime routines in families with young children; a DELPHI process for reaching wider consensus 

Dear Dr. Kitsaras:

I'm pleased to inform you that your manuscript has been deemed suitable for publication in PLOS ONE. Congratulations! Your manuscript is now with our production department. 

Kind regards, 

on behalf of

Dr. Srinivas Goli 

Academic Editor

PLOS ONE